# ATR Inhibitors in Platinum-Resistant Ovarian Cancer

**DOI:** 10.3390/cancers14235902

**Published:** 2022-11-29

**Authors:** Siyu Li, Tao Wang, Xichang Fei, Mingjun Zhang

**Affiliations:** 1Department of Medical Oncology, The Second Affiliated Hospital of Anhui Medical University, Hefei 230031, China; 2Department of Oncology, Anhui Medical University, Hefei 230031, China

**Keywords:** platinum-resistant ovarian cancer, targeted therapy, ATRi, PARPis

## Abstract

**Simple Summary:**

Platinum-resistant ovarian cancer (PROC) is a deadly cancer with a poor prognosis. Some drugs targeting ATR have shown initial success in the treatment of PROC. Therefore, we reviewed the mechanism of the ATR pathway, the results of preclinical and clinical trials in PROC, and potentially susceptible patients to ATR inhibitors. This study provides a basis for future experimental design and research.

**Abstract:**

Platinum-resistant ovarian cancer (PROC) is one of the deadliest types of epithelial ovarian cancer, and it is associated with a poor prognosis as the median overall survival (OS) is less than 12 months. Targeted therapy is a popular emerging treatment method. Several targeted therapies, including those using bevacizumab and poly (ADP-ribose) polymerase inhibitor (PARPi), have been used to treat PROC. Ataxia telangiectasia and RAD3-Related Protein Kinase inhibitors (ATRi) have attracted attention as a promising class of targeted drugs that can regulate the cell cycle and influence homologous recombination (HR) repair. In recent years, many preclinical and clinical studies have demonstrated the efficacy of ATRis in PROC. This review focuses on the anticancer mechanism of ATRis and the progress of research on ATRis for PROC.

## 1. Introduction

High-grade serous ovarian cancer (HGSOC) is one of the most fatal malignancies of the genitourinary system, and it poses hazards to the health of women [1]. Nearly 80% of patients with HGSOC were diagnosed with advanced disease, and the 5-year survival rate was approximately 32.1% [2,3]. The standard treatments for HGSOC include debulking surgery and platinum-based chemotherapy [4]. Most patients with HGSOC benefit from platinum-based chemotherapy after tumor recurrence. Nonetheless, many patients either relapse within 6 months or do not respond to platinum-based chemotherapy and are called “platinum-resistant/refractory”. Patients with platinum-resistant ovarian cancer (PROC) have a poor prognosis, with the median survival of platinum-sensitive patients being approximately 2 years and that of platinum-resistant patients being less than 12 months [5]. However, there are a limited number of treatments for PROC.

Due to the poor efficacy of existing treatments, new treatment strategies need to be developed. Targeted therapy currently plays an important role in tumor therapy, specifically in PROC [5]. In recent years, drugs designed to target DNA damage repair (DDR)-related pathways have attracted attention. Ataxia telangiectasia and RAD3-related protein kinase (ATR)/checkpoint kinase 1 (CHK1) have attracted great attention as possible targets for anticancer therapy because of their role in regulating cell cycle checkpoints. ATR inhibitors (ATRis), which target the ATR pathway, have also shown preliminary efficacy for the treatment of ovarian cancer (OC). Studies from the past decade on ATRi for treating PROC will be discussed here.

## 2. What Is PROC?

Traditionally, patients with OC who had a recurrence or disease progression within 6 months of their last platinum-based chemotherapy were defined as PROC patients; otherwise, they were defined as platinum-sensitive ovarian cancer (PSOC) patients. Later, the concept of “progression-free survival” was used to complement the definition of PROC. In 1989 and 1991, two retrospective analyses proposed the concept of the cisplatin-free interval as a factor influencing the response to treatment after OC recurrence [6,7], while Gore ME [8] used the progression-free interval as a prognostic factor for OC in 1990. Although the progression-free interval and the platinum-free interval have long been used synonymously, they are not the same. The platinum-free interval refers to the period from the end of primary platinum-based chemotherapy to disease progression or recurrence, whereas the progression-free interval includes the period of previous treatment. Based on the above three studies, OC is classified as “PSOC” or “PROC” according to whether the progression-free interval or platinum-free interval is longer than 6 months. In addition, patients who show progress during therapy or within 4 weeks of the last dose are defined as having “platinum-refractory ovarian cancer”. The broad definition of PROC would include platinum-refractory ovarian cancer.

Imaging lesions are the first factor that is considered when evaluating OC progression or recurrence. Additionally, the role of cancer antigen CA-125 elevation in the assessment should be carefully assessed. Both the progression-free interval and platinum-free interval define recurrence or progression according to Response Evaluation Criteria in Solid Tumor (RECIST) [9,10]. In 2000, the Gynecologic Cancer Intergroup expanded the definition of recurrence or progression of OC based on changes in the CA125 concentrations [11]. Based on the progression or recurrence of CA125 criteria and considering that maintenance therapy may have an impact on the sensitivity of patients to platinum, experts at the 4th Ovarian Cancer Consensus Conference of the Gynecologic Cancer Intergroup suggested that CA125 should be part of the platinum sensitivity evaluation criteria in OC. It should be noted that in assessing the date of progression, RECIST should be prioritized [12]. Moreover, the platinum-free interval or progression-free survival evaluated under the two distinct criteria should not be compared together.

It is essential to improve our understanding of the context of targeted therapies. Platinum compounds can bind to DNA to form intrastrand or interstrand DNA cross-links, inducing DNA damage reactions [13]. On-target resistance mainly involves the recognition and repair of DNA adducts and the subsequent impairment of apoptotic signaling pathways. For example, enhancement of homologous recombination (HR) is one type of that [14]. Hence, inhibition of DDR is a mechanism to overcome platinum resistance.

## 3. Therapeutic Strategies for PROC

The treatment strategy for PROC is a comprehensive treatment dominated by chemotherapy, targeted therapy, and symptomatic therapy. Besides, immunotherapy and surgery are also promising options. Single nonplatinum–based agents used sequentially are preferred for PROC. The response rate is 22% for docetaxel and 21% for weekly paclitaxel.

In addition, targeted therapy is involved in the treatment of PROC especially bevacizumab and poly (ADP-ribose) polymerases inhibitors (PARPis). The AURELIA III trials confirmed the efficacy of bevacizumab in PROC [15,16]. Overall, the response rate for bevacizumab alone is approximately 20% [17], which highlights the effectiveness of PARPis. In a phase II trial, olaparib had a higher ORR in patients with BRCA-mutated platinum-sensitive OC than in platinum-resistant patients (40% vs. 33%), and similar results were found in patients without BRCA mutations (50% vs. 4%). We note that this also means that olaparib is not completely ineffective against PROC [18]. In the ARIEL2 trial, rucaparib treatment had an ORR of 25% (5/20; 95% confidence interval (CI): 9–49) in patients with germline BRCA1/2 mutations [19].

Recently, immune checkpoint blockade (ICB) has been increasingly used in cancer treatment over the past few years. ICBs mainly involve programmed death 1 (PD-1)/programmed death ligand 1 (PD-L1) and anti-cytotoxic T lymphocyte antigen 4. Although immunotherapy is not currently a routine treatment for OC, it has shown a comparable response rate of 10% to 15% in monotherapy trials. In clinical trials, subgroups with BRCA mutations benefited more from immunotherapy combined with chemotherapy or PARPis [20].

Surgery is also a promising option for PROC, and the surgical methods for PROC mainly include secondary cytoreductive surgery, intraperitoneal hyperthermic chemotherapy, and palliative surgery. There are two retrospective analyses that showed that patients receiving secondary cytoreductive surgery plus chemotherapy had significantly longer postrelapse survival than patients treated with chemotherapy alone [21,22]. However, the addition of intraperitoneal hyperthermic chemotherapy to secondary cytoreductive surgery is still controversial. Palliative surgery is one form of palliative care. When patients suffer from life-threatening conditions such as intestinal obstruction that cannot be relieved medically, palliative surgery can relieve symptoms and alleviate suffering. In general, it is important to fully evaluate the patient’s general condition and indications for reoperation in patients with PROC because of the high postoperative mortality and perioperative complication rate [23].

Radiotherapy is also a form of palliative care for PROC. Historically, whole-abdominal radiotherapy was an option for early-stage and minimally residual advanced-stage ovarian carcinoma. Since the 1980s, platinum-based systemic chemotherapy has gradually replaced whole-abdominal radiotherapy for the management of OC. In recent years, radiotherapy has been used to control symptoms, improve tolerance, and increase the efficacy of chemotherapy, targeted therapy, and immunotherapy in patients with advanced OC, especially those with chemotherapy-resistant OC [24].

Doctors from multiple disciplines are involved in the treatment of PROC. Therefore, multidisciplinary collaboration can lead to better outcomes for patients. Multidisciplinary collaboration is a better strategy to manage patients with chronic diseases such as cancer. In multidisciplinary teams, doctors from different departments share information and treatment options in a patient-centered manner and ultimately develop personalized therapeutic options for the patient [25].

## 4. ATR/CHK1 Pathway

ATR is a 2660 amino acid-long mammalian lineal homolog of mitotic entry checkpoint proteins in yeast, that was discovered in 1996 [26]. It is an important member of the phosphatidylinositol 3-kinase-related kinase (PI3K) family and regulates the cell cycle by responding to single-stranded DNA (ssDNA) breaks, especially those caused by replication stress (RS) [27]. Targeting the ATR/CHK1 pathway provides an opportunity to prevent unrepaired double-stranded breaks (DSBs) and under-replicated DNA from being involved in mitosis, which would result in mitotic catastrophe and cell death.

The accumulation of ssDNA caused by various factors initiates the ATR/CHK1 pathway. Due to various endogenous and exogenous factors, such as ultraviolet radiation (UV), ionizing radiation (IR), chemical agents, and spontaneous damage in DNA replication, various forms of damage occur in the human DNA, among which ssDNA is the most common [28]. Cells rely on the DDR to repair DNA damage and thereby avoid cell death. While DDR activates cell cycle checkpoints to provide time to repair damaged DNA, it also directly repairs DNA by regulating gene expression [29]. DDR blocks the cell cycle and produces more ssDNA, which is covered by replication protein A (RPA) to maintain single-stranded structural stability [30,31]. RPA recruits ATR and the ATR-interacting protein (ATRIP) to foci with DNA damage, and stimulates the binding of ATRIP and ssDNA [32].

ATR acts as a sensor that activates CHK2 after sensing ssDNA or RS, leading to cell cycle arrest (Figure 1). The ATR kinase substrate is recruited as a scaffold based on a proliferating cell nuclear antigen (PCNA)-like trimer complex composed of Rad9-Rad1-Hus1 (9-1-1) [33]. The ATRIP-ATR complex phosphorylates 9-1-1, which recruits topoisomerase-binding protein 1 (TOPBP1). TOPB1 then binds to the PI3K regulatory domain of the ATRIP complex to activate ATR, triggering the ATR cascade signaling pathway [27,34,35]. Activated ATR phosphorylates CHK1 at Ser317 and Ser345; activated CHK1 then inactivates CDC25A, CDC25B, and CDC25C through phosphorylation. CDC25A is responsible for activating CDK2 through substrate dephosphorylation and promotion of G1 phase progression. CDC25C protects CDK1 from 14-3-3 proteins by promoting the binding of 14-3-3 proteins to enzymes [34,35,36]. CHK1 can also phosphorylate and stabilize WEE1 [37]. Ultimately, decreased activation of CDK1/cyclin B kinase results in G2/M stagnation, allowing time for DNA repair.

In addition, ATR can promote HR [38,39] and RAD51 function [40]^.^ either directly or through CHK1 [41,42]. And ATR can also recruit BRCA2 and RAD51 to sites of DNA damage [43]. Short-term (less than 24 h) inhibition of ATR decreases RAD51 phosphorylation and impairs ATR-mediated protein-protein interactions such as BRCA1 interactions with partner and localizer of breast cancer 2 (PALB2) and TOPBP1. While long-term (5–8 days) treatment can increase CHK1 -mediated activation of transcription of E2F to impair end resection, RAD51 and RAD52 foci formation reduces the abundance of HR proteins including BRCA1 and FANCJ. Moreover, the ATR/CHK1 pathway plays a very important role in stabilizing the replication fork [44,45,46]. Due to its important role in cell cycle regulation and DNA repair, ATR has been considered an important target for anticancer therapy.

## 5. Preclinical Trials of ATRi

Proliferation driven by oncogenes or the administration of radiotherapy and chemotherapy can lead to increased RS in cancer cells. The repair of damage caused by RS depends on blocking the cell cycle via cell cycle checkpoints. ATRis promote elongation of the G2/M phase and reduce the amount of HR, resulting in the accumulation of intracellular DNA damage and ultimately apoptosis [29]. As mentioned above, the accumulation of ssDNA triggers the ATR pathway. The therapeutic effects of ATRi are based on targeting the DNA damage caused by radiotherapy or chemotherapy. Therefore, ATRis have been studied in combination therapies for OC because of the potential for greater benefits for patients. Herein, we will sunmmarize the current studies on the combination of classical treatments and ATR inhibitors for PROC.

### 5.1. Combination with Radiotherapy

IR has been used extensively to treat tumors because of its ability to induce DNA damage, which can result in direct tumor cell death. ATRis initially attracted attention as radiosensitizers. Moreover, ATRis can increase the therapeutic ratio (lethal dose of tumor tissues/tolerated dose of normal tissues), which leads to better efficacy and fewer adverse effects.

Schisandrin B is described as the first ATR-selective inhibitor, leading to sensitization of tumor cells in response to UV irradiation [47]. In 2012, Fokas et al. first reported that the addition of VE-822, the first intensively studied ATR inhibitor, increased γH2AX phosphorylation and the persistence of DNA damage caused by radiation in pancreatic ductal adenocarcinoma (PDAC) in vivo and in xenografts. This confirmed the radiosensitization of VE-822. The study also showed that VE-822 was not highly toxic to normal cells and tissues. This means that ATRi can increase the tolerated dose in normal tissues in the target area while reducing the lethal dose in tumor tissues, i.e., improving the therapeutic ratio [48]. The study by Teng et al. [49] confirmed that VE-821 (the former of VE-822) can enhance the reactivity to platinum drugs and the response to IR in several gynecologic cell lines, including OC cell lines. IR-induced DNA damage can activate the ataxia–telangiectasia mutated (ATM) and ATR signaling pathways. Complete loss of basally phosphorylated CHK1 levels was observed in cells treated with ATRi and IR compared to cells exposed to IR alone or treated with IR and an ATM inhibitor, independent of p53 status. These results highlight opportunities to enhance the efficacy of radiotherapy for PROC.

### 5.2. Combination with Chemotherapy

ATR inhibitors can not only make platinum-resistant cells respond to platinum, but also make them more sensitive to other chemotherapy drugs.

Several studies have shown that ATRis can reverse the resistance of cancer cells to platinum. Studies have shown that disruption of ATR function through depletion or kinase-dead protein expression could influence the survival of several cancer cell lines, including osteosarcoma cancer cells, lung cancer cells, and colon cancer cells, with or without DNA-damaging agents. More importantly, the tumor cell killing effect was more pronounced when combined with other chemotherapy agents, the most significant of which were the cross-linking drugs cisplatin and carboplatin [50]. This finding provides a potential strategy for platinum-based compounds in the treatment of PROC. In a study by Hall et al. [51], the addition of VX-970 significantly increased the sensitivity to platinum in vitro in non-small cell lung cancer that was previously insensitive to platinum. Moreover, VX-970 had the most obvious synergistic effect on the cells with the lowest initial platinum response. Platinum and VX-970 also showed a significant synergistic effect in a xenograft model. A study showed depletion of ATR with siRNAs that sensitized OVCAR-8 cells, a kind of OC cell line, to platinum, topotecan, and veliparib exposure. Similar results were obtained when the researchers used VE-821 to inhibit ATR. Moreover, ATRi might sensitize cells by altering the phosphorylation of other currently unclear substrates rather than Chk1 Ser345 [52]. Another study showed that inhibition of the ATR/CHK1 pathway reversed CXCL2-mediated platinum resistance in HGSOC cells [53].

Meanwhile, studies have also shown that ATRis broadly sensitize cancer cells to gemcitabine [54], paclitaxel [55], and doxorubicin [56]. These results suggest that ATRis could be used to treat PROC.

### 5.3. Combination with Immunotherapy

ATRis downregulates the expression of PD-1/PD-L1 and increases NK cell and T-cell lethality in the tumor microenvironment. This makes it possible that ATRis could synergistically enhance the anticancer efficacy of immunotherapy.

Multiple studies have shown that PD-L1 upregulation in cancer cells is ATM/ATR/Chk1-dependent and induced by IR or treatment with some DNA-damaging agents. Patients with genome instability, such as microsatellite instability (MSI), mismatch-repair (MMR)- deficiency, and HR deficiency, are more responsive to ICB [57]. Genomic instability is also an element of ATRi sensitivity, as detailed below. This means that people with this genome instability may benefit more from combination therapy, which can be used as a screening or grouping tool for patients recruited in clinical trials. In 2018, Sun et al. [58] demonstrated that an ATR inhibitor destabilizes the PD-L1 protein in a proteasome-dependent manner and attenuates the PD-L1/PD-1 interaction, resulting in increased sensitivity to T-cell killing. Another study showed that the combination of radiation and ATRi therapy boosted NK cell activity in the tumor microenvironment [59]. This provides the basis for the ICBs to be used in combination with ATRi.

### 5.4. Combination with Drugs That Target DNA Damage Repair Pathways

Multiple drugs that target DDR pathways combined with ATRi can synergistically induce cell death. When DNA damage occurs, ATR can activate the DDR pathway and block cell cycle progression to allow time for DDR and ensure cell survival or apoptosis [60]. In this case, the cells with defective DDR will carry a large amount of damaged DNA into the mitosis phase, which eventually leads to synthetic lethality. Synthetic lethality is defined as inactivating one of the two genes (or pathways) alone without affecting cell survival however, simultaneous inactivation of another gene results in cell death [29]. Here we focus on the targets that are better candidates for research.

#### 5.4.1. ATRi and CHK1 Inhibitor, ATM/CHK2 Axis Inhibitor

Although there is evidence that ATRis have a synergistic effect with CHK1 inhibitors and ATM inhibitors, they have not yet been subjected to clinical trials for various reasons, such as side effects or poor efficacy.

CHK1 is a member of the ATR/CHK1 pathway. Andrew J. Massey examined the interaction between V158411 (CHK1i) and VX-970 (ATRi). The results showed that the ATRi boosts the Chk1 inhibitor’s induction of DNA damage, RS, and tumor cell growth. This suggests that combining ATRi and Chk1 inhibitors may be a useful clinical approach in a wide range of cancers [61]. However, it is not clear whether this combination will result in better clinical efficacy and more side effects, as studies are also underway with CHK1 inhibitors.

ATM belongs to the PI3K kinase family as an ATR. The ATM/CHK2 pathway regulates the G1/S cell cycle in response to DNA DSBs. A study has shown that defects in the ATM pathway sensitize cells to ATRis plus cisplatin. Compared with normal ATM cells, ATM-deficient cells had lower S/G2 arrest and a significantly increased amount of γH2AX phosphorylation. These cells were also more sensitive to ATRis plus cisplatin [50]. In the study by Pang-ning Teng et al., they found that although the combination of an ATM inhibitor and an ATRi could synergistically enhance the reactivity of gynecological tumor cells to IR, the combination did not synergistically sensitize cells to platinum [49]. In addition, ATM inhibitors have not gone far as treatments for tumors due to side effects and various other reasons. Hence, ATM has been studied more as a molecular marker than as a therapeutic target.

#### 5.4.2. ATRi and Poly (ADP-Ribose) Polymerase Inhibitors (PARPi)

Due to the effect of a poly (ADP-ribose) polymerase inhibitor (PARPi) on HR, the synergistic anticancer effect of ATRis and PARPis is attracting wide attention, especially in PROC with the BRCA mutation.

The ATR/CHK pathway plays a role in acquired PARPi resistance, which provides a rationale for the combination of PARPis and ATRis for therpy. Poly (ADP-ribose) polymerases (PARPs) are enzymes that mediate base excision repair of ssDNA damage. Inhibition of PARP1/2 induces synthetic lethality in cells with defective DNA repair mechanisms, such as BRCA mutations or homologous recombination deficiency (HRD) [62,63]. Although PARPis play a dominant role in HGSOC, unfortunately, there is resistance in PROC. PARPi resistance may emerge in a variety of mechanisms in PROC, the most common of which is restoration of HR capacity [64]. As described earlier, inhibiting ATR/CHK1 can suppress HR, which may allow patients to overcome PARPi resistance.

Multiple studies have shown that combination therapy with PARPi and ATRi, especially long-term ATRi, is very promising. In 2013, Huntoon et al. [52] reported that VE-821 significantly enhanced the sensitivity of OC cell lines with BRCA deficiency to veliparib (a kind of PARPi). Thereafter Kim et al. [65] demonstrated the synergistic effect of AZD6738, a kind of ATRi, combined with PARPis in HGSOC both in vivo and in vitro. The use of PARPis increased the ATR/CHK1 pathway dependence in HGSOC with mutated BRCA. The combination of PARPis and ATRis resulted in cell retention in the G2-M phase and increased DNA damage and apoptosis. A similar study in 2020 in platinum-resistant or PARPi-resistant OC cells showed that treatment of cells with PARPis increased CHK1 phosphorylation and stagnated G2/M cycle cell growth, and the addition of ATRis reversed these effects. In addition, the colony formation ability of the cell decreased significantly after ATRi-PARPi treatment compared with PARPi treatment alone. To explore the synergistic mechanism of ATRis and PARPis, the accumulation of DNA DSBs in the S phase and the changes in copy fork and apoptotic markers were evaluated after combination treatment. Overall, the results suggested that ATRi-PARPi may increase DNA DSBs in the S phase by increasing RS, ultimately leading to platinum-resistant or PARPi-resistant OC cell death through apoptosis. In mouse PDX models of platinum-resistant and PARPi-resistant OC with BRCA1 mutation or *CCNE1* amplification, ATRi-PARPi combination therapy had considerable efficacy and safety [66]. This result provides more evidence for PROC treated with ATRi-PARPi. Dibitetto et al. [44] showed that long-term pretreatment with ATRis selectively induced hypersensitivity of cancer cells to PARPis because of oncogene-induced high RS without interfering with the reactivity of normal cells to PARPis. In the other study [46], researchers found that using ATRis and PARPis after a low-dose, long-term (five days) pretreatment with ATRis to deplete key HR components made cancer cells particularly sensitive to PARPis. This indicates a more likely way to benefit from combination therapy with ATRis and PARPis as follows: a low dose of ATRi pretreatment followed by long-term combination with a PARPi.

Although the performance of PARPis was worse in PROC patients than in PSOC patients, the treatment still had a satisfactory outcome [19,67]. The NCCN guidelines, therefore, recommend rucaparib as a first-choice recommended treatment for PROC when there are few better options [4]. PARPis play such an important role that overcoming its resistance by ATRi combination may be beneficial for PROC patients.

#### 5.4.3. ATRi and Topoisomerase Inhibitors

ATRi can improve the sensitivity of a variety of cancer cells to DNA topoisomerase (Topos) inhibitors. Treatment using AZD6738 administered intermittently with Topos inhibitors should be a priority to study.

DNA Topos plays essential roles in the unwinding and disentanglement of DNA during replication and transcription. Topos are classified as either topoisomerase type I (Top1) or type II (Top2) based on function, reaction mechanism, and structure [68]. The Topos inhibitors topotecan and irinotecan are preferred for PROC, while studies have shown that the combination therapy of ATRis and specific Top1 inhibitors such as topotecan, irinotecan [69], and belotecan [70] can promote the sensitization of chemotherapy-resistant OCs to chemotherapy.

In 2014, Jossé et al. [69] first selected ATR as the preferred synthetic lethal gene for Top1 inhibitors by siRNA screening. Knockdown of ATR sensitized breast carcinoma cells to Top1 inhibitors, which further supported this conclusion. VE-821 sensitizes breast and colorectal cancer cells to Top1 inhibitors, but it has no remarkable impact on apoptosis when used as a monotherapy. The combined antitumor conspicuous effect of VE-822 in vivo and irinotecan on colorectal cancer in vivo was also confirmed. In 2021, Hur et al. [70] found that ATR inhibition by the addition of AZD6738 to belotecan (Top1 inhibitor) dephosphorylates Thr14 and Thy15 of CDK1 to relieve cell cycle arrest, leading to mitotic catastrophe and cell apoptosis in paclitaxel-resistant OC. Moreover, AZD6738 and belotecan synergistically inhibited the proliferation of PROC derived from the ascitic fluid samples of OC patients. However, AZD6738 was not tolerated in a nude mouse model using 30 or 40 mg/kg QD plus belotecan 20 mg/kg (every 4 days) or belotecan 10 mg/kg (every 4 days). The intermittent administration of this combination regimen was tolerated. The therapeutic prospects of ATRis and Topos inhibitors in PROC are promising.

#### 5.4.4. ATRi and WEE1 Inhibitors

The synergistic antitumor effect of WEE1 inhibitors and ATRis is definite, and more trials are needed to verify their efficacy and tolerability in vivo.

WEE1 is another key required for cell cycle checkpoint arrest in the S/G2 phase. Furthermore, WEE1 inhibition causes unplanned triggering of replication and activates endonucleases in the S-phase, leading to the accumulation of damaged DNA [71]. Studies have shown that the combination of WEE1 inhibitors and ATRis has a synergistic effect on lung cancer cells [72] and multiple OC cell lines, including OVCAR3, SKOV3, OV90, OVCAR8, and A2780. These studies demonstrate that the WEE1 inhibitor-ATRi combination induces an immune response not only by regulating the cell cycle and HR but also by activating the STING signaling pathway, enhancing the type I interferon (IFN-IS) response, acting on tumor-infiltrating lymphocytes, and upregulating PD-L1 expression. In a subsequent mouse model, the addition of a PD-L1 antibody enhanced the efficacy of ATRi-WEE1i in the treatment of colorectal cancer [73]. In another study, ATRi-WEE1i was shown to reduce the viability and colony-forming capacity of CCNE1-amplified OC cells [74].

## 6. Clinical Trials of ATRi

Numerous cell- or animal-model-based test studies provide a solid theoretical basis for clinical trials of ATRis. As of the date of this report (1 September 2022), many ATRi-related clinical trials have been registered and made available, of which 48 accepted patients with solid tumors including advanced OC, and 15 focused specifically on OC (Table 1).

### 6.1. Berzosertib

Berzosertib (previously referred to as VE-822, M6620, and VX-970) is the first high-selection ATRi for injection to enter clinical trials. It inhibited the ATR-dependent phosphorylation of γH2AX, which altered the recognition and repair of DNA DSBs, and subsequently led to the accumulation of DNA damage [51]. We used berzosertib, VE-822, M6620, and VX970 as keywords to search Pubmed, the Cochrane Library, the Wiley Online Library, and the ELSEVIER ScienceDirect Database and found a total of 15 results on clinical trials of berzosertib.

On December 10, 2012, a phase I clinical trial of berzosertib was started. This clinical study demonstrated that berzosertib was well tolerated with topotecan in advanced solid tumors (topotecan 1.25 mg/m^2^, days 1 to 5; M6620 210 mg/m^2^, days 2 and 5) and achieved satisfactory efficacy in platinum-resistant SCLC (3 of 5 derived a durable clinical benefit) [75]. Multiple Phase I trials showed that the recommended phase II dose (RP2D) of berzosertib was 210 mg/m^2^ with intermittent use of other drugs [76,77,78].

We found the results of 3 published phase II clinical trials of berzosertib in OC. In phase II trials, the combination of berzosertib (210 mg/m^2^) on day 2 and day 9 and gemcitabine (1000 mg/m^2^) on day 1 and day 8 of a 21-day cycle showed better anticancer efficacy than gemcitabine alone in platinum-resistant HGSOC (the median OS: 59.4 weeks (90% CI 33.7–84.4) vs. 43.0 weeks (34.4–67.9); hazard ratio 0.84, 0.53–1.32; one-sided log-rank test *p* = 0.26), especially in patients with platinum-free intervals of less than 3 months (84.4 weeks (59.4–unreached) vs. 40.4 weeks (27.6–92.4); hazard ratio: 0.42, 0.19–0.94; one-sided log-rank test *p* = 0.034). In addition, the adverse reactions of grade 3 or greater in the combination therapy group were similar to those in the monotherapy group, of which neutropenia and thrombocytopenia were the most common. It should be noted that in the combined treatment group, 2 of 34 patients stopped taking the drug due to pneumonia, and 1 person died of pneumonia. Grade 2 pneumonia occurred in 1 of 36 patients in the gemcitabine monotherapy group [79,80]. Due to the small sample size and the ability of gemcitabine to cause interstitial pneumonia [81], it is difficult to determine whether berzosertib causes pneumonia or if it induces or aggravates gemcitabine-induced pneumonia. Regardless, pneumonia will be one of the most important side effects of berzosertib to consider in future clinical trials. In addition, the phase II trial of berzosertib combined with avelumab and carboplatin in PARPi-resistant PSOC was terminated by the investor after the end of the phase Ib trial (ClinicalTrials identifier: NCT03780608) [82].

### 6.2. Ceralasertib

Ceralasertib (AZD6738) is an oral ATRi with the most trials underway. It can inhibit the phosphorylation of CHK1 and increase the phosphorylation of γH2AX [83]. We found 17 results on clinical trials of berzosertib, and three of them were related to OC.

The RP2D of berzosertib monotherapy in all Phase I trials ranged from 160 mg ququaque die (QD) on days 1–7 to 240 mg bis in die (BID) on days 1–14 in 28-day cycles [84,85,86]. However, a phase I trial of ceralasertib combined with carboplatin in advanced solid tumors was not extended to phase II due to the side effects of myelotoxicity, especially neutropenia and thrombocytopenia. Considering that the dosage of ceralasertib (20 mg BID on days 4–13 and 40 mg QD on days 1–2) is obviously lower than that of RP2D of other trials and the fact that all the enrolled patients have undergone multiline chemotherapy and their bone marrow function has been damaged, these results do not mean that ceralasertib cannot be used in combination treatment with carboplatin at all [87].

CAPRI [88] is a phase II trial in which 14 patients with platinum-resistant HGSOC were enrolled in cohort B to receive ceralasertib in combination with olaparib. Grades 4–5 adverse events did not occur in this trial. Three patients showed a response to CA-125 and one patient showed improvement in symptoms; however, none of the patients achieved a response by RECIST V1.1. The OLAPCO trial [89] enrolled patients with DDR-related alterations. Patients with germline and somatic mutations treated with ceralasertib in combination with olaparib achieved an 8.3% ORR and a 62.5% clinical benefit rate. Two of five patients with ATM mutations achieved complete responses (CR) or stable disease (SD) for more than 24 months. The enrolled patients included seven PARPi-resistant HGSOC patients with BRCA mutations. One of these seven patients achieved a partial response (PR), five achieved a SD, and the overall clinical benefit rate was 85.7%. Importantly, patients benefited from the ceralasertib plus olaparib regimen for longer than they did when they first received PARPis (median duration of benefit: 8 months (range 2–18) months v 4 months (range 2–12)). The most common side effect was hematologic toxicity. Interestingly, responses were also observed in patients given reduced doses because of adverse events. The combination of ceralasertib and durvalumab, an anti-PD-L1 antibody, showed good activity and tolerable toxicity against melanoma [90], while it had failed against prior anti-PD-L1 drugs and advanced chemotherapy-refractory gastric cancer [91] in phase II clinical trials. Subsequent biomarker analysis revealed that patients with altered DDR pathways responded better to combination therapy [90].

### 6.3. M1774

M1774 is an orally administered ATRi. Only one clinical trial has been published thus far. The data in this trial showed that M1774 monotherapy in patients with advanced solid tumors is valid and well tolerated. The reduction in γH2AX levels in target and peripheral blood mononuclear cells was shown at a dose of 130 mg QD [92].

More clinical studies with other ATRis are ongoing.

## 7. Molecular Markers and Determinants of ATRi Sensitivity

It is crucial to identify patients who could benefit from ATRis for the design of clinical trials and the treatment with these new drugs. Several genetic alterations have been reported to be associated with the sensitivity of cancer to ATRis. We classified the mechanisms of molecular markers that sensitize cells to ATRis into four categories (Table 2). It should be pointed out that the mechanism of synthesizing lethality between each marker and ATRi is not singular.

The blocking of DNA synthesis and the accumulation of damage mainly refer to the failure of DNA synthesis caused by a defect of DNA synthetase or spontaneous DNA damage due to a defect of the marker. For example, defects in *POLD1*, *POLE3/POLE4, REV3,* and PRIM1 lead to defects in DNA polymerases DNA polymerase (Pol) δ [102], Pol ε [101], Pol ζ [105], and DNA primase polypeptide 1 [104], respectively. In addition, *RNASEH2* deficiency [106] ultimately leads to impaired DNA synthesis and spontaneous DNA damage. Cells with all these defects are more susceptible to ATRis.

As mentioned above, ATR plays an important role in DDR. Therefore, several studies have shown that cells deficient in DDR, particularly HR-related pathways, are sensitive to ATRis. HR is an error-free form of the DDR, which uses a sister chromatid as a template. In cancers with HRD, non-homologous end-joining (NHEJ) will repair DNA DSBs with lower fidelity as an alternative, which results in genomic instability [134]. Thus, most of the defects in HR pathways cause cells to be more sensitive to ATRis. Because of the importance of BRCA mutations in OC, the effect of BRCA mutations on ATRi sensitivity is highlighted here. Compared with the BRCA wild-type OC cell line, ATRis significantly decreased cell viability in the BRCA-mutated OC cell line [108]. In addition, O^6^-methylguanine-DNA methyltransferase (MGMT) and DNA-PKCs are important proteins in the direct repair and NHEJ repair pathways, respectively. Both ERCC1 and XRCC1 play important roles in base excision repair pathways. PTEN is involved in several DDR pathways [135]. Defects in these proteins can also lead to greater sensitivity to ATRis.

RS was defined as the slowing or stalling of replication fork progression and/or DNA synthesis [136]. Increased RS promotes ATR activation. Bradbury et al. [137]. demonstrated that increased RS confers sensitivity to ATRis. Several genetic mutations cause alterations in RS elevation that also lead to sensitivity to ATRis, such as *BRG1* loss (*SMARCA4*-deficient) [96], reduced APOBEC3B [137], MYCN amplification [123], and so on.

Cell cycle checkpoints are mainly induced by the p53, ataxia-telangiectasia-mutated (ATM)/CHK2, ATR/CHK1, and P38/MK2 pathways. Therefore, we hypothesized that cells with mutations in other cell cycle checkpoints tend to be more dependent on ATR and thus more sensitive to ATRis. In vitro and in vivo experiments have also confirmed that ATM loss increases sensitivity to ATRis in various types of cancer [138,139]. In cells with mutations in TP53, however, things are not so simple. Although ATRi can increase RS in p53-deficient cells, it does not cause them to be more sensitive to ATRi than genetically matched p53-wild-type cells. Middleton et al. [140] speculated that the effectiveness of ATRis may be more dependent on tumor type than p53 status. Cyclin E1 protein binds to CDK2 to take the cell cycle from the G1 phase to the S phase, whereby cells with CCNE1 amplification became more dependent on the G2-M checkpoint [141]. This may be the reason why CCNE1-amplified cells are more sensitive to ATRi-WEE1i and ATRi-PARPi combinations [74]. Furthermore, some proteins, such as nucleolar and spindle-associated protein 1 (NUSAP1) [131], Tim-Tipin [132], and WWOX [133], influence the ATR pathway, whereby cells deficient in these markers are more sensitive to ATRi.

Approximately 50% of HGSOC patients have mutations in HR-related genes, including BRCA1/2, RAD51, and ATM [142,143]. On the one hand, the absence of these checkpoints may be a hint that patients with OC are more likely to respond to ATRis. On the other hand, they also provide a basis for screening patients for clinical trials and clinical treatment.

## 8. Discussion

In this paper, we briefly introduce PROC and its treatment options. We also review the function of the ATR/CHK1 pathway and the preclinical and clinical results. Although some points are still worth exploring.

(1)If lowering the dose and extending the duration of medication increase its efficacy, that remains an open question. ATRis inhibit the activation of ATR and its downstream molecules, leading to the restart of the cell cycle and the damage to HR. It should be remembered that long-term inhibition of the ATR/CHK1 pathway is more damaging to HR by increasing the transcription of E2F. This implies that long-term, low-dose treatment seems to be a better option. However, the results of models in vitro and clinical trials indicate that the toxicity of continuous administration appears to be intolerable. It could also be that the combination with chemotherapeutic drugs adds toxicity. It is worth trying to lower the dose or extend the treatment time in future experiments. Another option would be to pretreat with ATRis before using other chemotherapeutic agents or targeted agents.(2)What types of cancers should be selected in the design of in vitro experiments with ATRis? We found that ATRis also showed anticancer effects in chronic lymphocytic leukaemia, mantle cell lymphoma, non-small cell lung cancer, gastric cancer and other tumors [138,139,144,145]. Middleton et al. [140] speculated that tumor benefit from ATRis may be more dependent on tumor type. It is crucial to carefully select the types of cell lines used in studies. We hypothesize that cancers with a high RS or a high incidence rate of defects in DDR might be a good choice.(3)Is there an opportunity for ATR inhibitors to expand the indication for PARPis? As an important treatment option for OC, PARPis are now widely used in patients with HRD and PROC. As pointed out in our literature review, ATRis can overcome cells’ resistance to PARPis, enhancing the response to PARPis. This makes it possible for PARPis to perform better in PROC patients. However, it is necessary to test this in more cell lines with initial resistance to PARPis.(4)How effective are ATRis in other pathological types of OC? As described above, ATRis have a common chemotherapy sensitization effect in gynecological tumor cell lines, independent of BRCA status. As ATRi clinical studies are still mainly in phase II, almost all enrolled patients were advanced cancer patients who were resistant to platinum after multiline therapy. In addition, the efficacy of ATRis in patients with other types of OC is unclear due to the higher incidence of HGSOC. Patients with other types of OC may also benefit from clinical trials of ATRis.

## 9. Conclusions

Overall, ATRis can serve as potential candidates for the targeted therapy of PROC, especially in combination with PARPis. High levels of RS or DDR-related gene mutations, such as BRCA1/2 or ATM mutations and CCNE1 amplification, may be markers of ATRi sensitivity. However, data from clinical studies with larger sample sizes is still needed. We expect that ATRis will bring good news for more OC patients in the future.

## Figures and Tables

**Figure 1 cancers-14-05902-f001:**
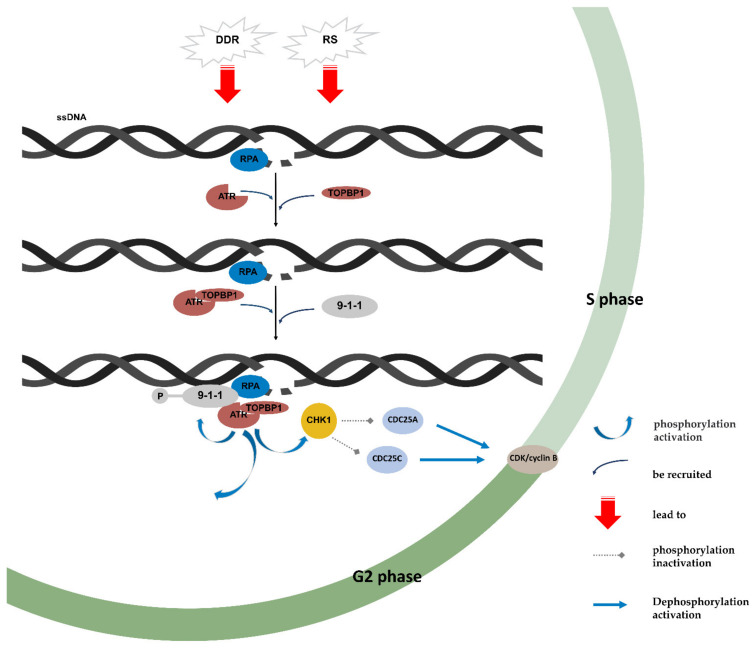
Mechanism of ATR/CHK1-mediated regulation of the cell cycle. Abbreviations: 9-1-1: Rad9-Rad1-Hus1; ATR: ataxia telangiectasia and RAD3-Related Protein Kinase; CDK: cyclin-dependent kinases; CHK1: checkpoint kinase 1; DDR: DNA damage reactions; RPA: replication protein A; RS: replication stress; ssDNA: single-stranded breaks DNA; TOPBP1: topoisomerase-binding protein 1.

**Table 1 cancers-14-05902-t001:** Summary of clinical trials related to ATRi.

ATRi	Intervention	Main ID	Phase	Status	Condition or Disease
ART0380	ART0380; ART0380 + Gemcitabine; ART0380 + Irinotecan	NCT04657068	I/II	Recruiting	OC, advanced cancer, metastatic cancer, primary peritoneal cancer, and fallopian tube cancer with DDR genes
ATRN-119	ATRN-119	NCT04905914	I/II	Recruiting	Advanced solid tumor with DDR genes
Berzosertib (VE-822, M6620, VX-970)	Berzosertib	NCT03718091	II	Completed	Solid tumor, leiomyosarcoma, and osteosarcoma with HR mutations
	Berzosertib + Avelumab	NCT04266912	I/II	Recruiting	DDR-deficient metastatic or unresectable solid tumors with DDR genes
	Berzosertib + Carboplatin	EUCTR2013-005100-34-GB	I	Not recruiting	Advanced-stage solid tumors with DDR genes
	Berzosertib + Carboplatin + Avelumab	NCT03704467	I	Completed	PARPi-resistant OC with BRCA 1/2 mutation
	Berzosertib + Carboplatin + Avelumab	EUCTR2018-001534-17-BE	Ib/II	Not recruiting	PARPi-resistant recurrent ovarian, primary [eritoneal, or fallopian tube cancer
	Berzosertib + Carboplatin + Paclitaxel	NCT03309150	I	Active	Advanced-stage solid tumors
	Berzosertib + Carboplatin + Gemcitabine hydrochloride	NCT02627443	I	Active	Platinu- sensitive recurrent and metastatic ovarian, primary peritoneal, or fallopian tube cancer
	Berzosertib + Cisplatin + Gemcitabine; Berzosertib + Cisplatin + Etoposide; Berzosertib + Irinotecan; Berzosertib+Gemcitabine; Berzosertib + Cisplatin; Berzosertib + Carboplatin	NCT02157792	I	Completed	Advanced-stage solid tumors
	Berzosertib+ Chemotherapy	EUCTR2012-003126-25-GB	I	Not recruiting	Advanced Solid Tumors
	Berzosertib + Gemcitabine hydrochloride	NCT02595892	II	Completed	Platinum-resistant recurrent ovarian, primary peritoneal, or fallopian tube cancer
	Berzosertib + Irinotecan Hydrochloride	NCT02595931	I	Recruiting	Solid tumors that are metastatic or cannot be removed by surgery with DDR genes
	Berzosertib+Lurbinectedin	NCT04802174	I/II	Recruiting	Advanced solid tumors, SCLCS, mall cell cancers, and high-grade neuroendocrine cancers
	Berzosertib + Topotecan	NCT05246111	I	Recruiting	Advanced solid tumor
	Berzosertib + Veliparib + Cisplatin	NCT02723864	I	Completed	Refractory solid tumors
Ceralasertib (AZD6738)	Ceralasertib	NCT04564027	II	Recruiting	Advanced solid tumours with deleterious ATM mutation
	Ceralasertib	EUCTR2020-002529-27-FR	IIa	Authorised	Advanced cancer whose tumours contain molecular alterations
	Ceralasertib; Ceralasertib+ Carboplatin; Ceralasertib + Olaparib; Ceralasertib+ Durvalumab	NCT02264678	I/II	Recruiting	Platinum sensitive OC with BRCA mutant or RAD51C/D mutant or HRD positive status, head and neck SCC, ATM proficiency/deficiency NSCLC, and gastric or breast cancer
	Ceralasertib; Ceralasertib + Olaparib; Adavosertib + Olaparib	NCT03579316	II	Recruiting	Recurrent ovarian, primary peritoneal, or fallopian tube cancer
	Ceralasertib; Ceralasertib + Olaparib	EUCTR2019-003791-39-GB	II	Authorised	platinum-sensitive epithelial ovarian cancer
	Ceralasertib+ Durvalumab	KCT0003806	II	Not recruiting	Metastatic solid tumor
	Ceralasertib+ Durvalumab	CTR20221743	I	Not recruiting	Advanced solid tumors
	Ceralasertib+ Gemcitabine	NCT03669601	I	Recruiting	Advanced or metastatic solid tumour
	Ceralasertib+ Gemcitabine	EUCTR2017-003935-12-GB	I	Authorised	Advanced or metastatic solid tumour
	Ceralasertib + Olaparib	NCT02576444	II	Active	Cancer, including HGSC harboring DDR, and repair alterations
	Ceralasertib + Olaparib	NCT03462342	II	Recruiting	Recurrent OC
	Ceralasertib + Olaparib	NCT03878095	II	Recruiting	Malignant solid neoplasm, refractory cholangiocarcinoma, or refractory malignant solid neoplasm with IDH1 and IDH2 mutant
	Ceralasertib monotherapy; Ceralasertib + Olaparib	NCT04065269	II	Recruiting	Gynaecological cancers
	Ceralasertib + Olaparib; Olaparib monotherapy	NCT04239014	II	Withdrawn	OC
	Ceralasertib + Paclitaxel	NCT02630199	I	Completed	Refractory cancer
	Ceralasertib + Paclitaxel	KCT0003403	I	Recruiting	Advanced or metastatic solid tumour
	Ceralasertib + Palliative Radiotherapy	NCT02223923	I	Unknown	Solid-tumour refractory to conventional treatment
Elimusertib (BAY1895344)	Elimusertib	NCT03188965	II	Active	Advanced solid tumor and lymphomas with DDR defects
	Elimusertib	NCT05071209	I/II	Recruiting	Relapsed or refractory solid tumors
	Elimusertib + Cisplatin; Elimusertib + Cisplatin + Gemcitabine	NCT04491942	I	Recruiting	Advanced solid tumors with emphasis on urothelial cancer
	Elimusertib + Copanlisib	NCT05010096	Ib	Withdrawn	Advanced solid tumors with at least one DRR-related gene mutation
	Elimusertib+ Gemcitabine	NCT04616534	I	Recruiting	Advanced solid tumors, advanced pancreatic and OC, and advanced solid tumors
	Elimusertib +Niraparib	NCT04267939	Ib	Recruiting	Recurrent EOC, fallopian tube, or primary peritoneal cancer, and recurrent advanced solid tumors
	Elimusertib+ Pembrolizumab	NCT04095273	I	Active	Advanced solid tumor with putative biomarkers of DDR deficiency
M1774	M1774; M1774 + Niraparib	NCT04170153	II	Recruiting	Metastatic or locally advanced unresectable solid tumors
	M1774+ DDR inhibitor; M1774 + Immune Checkpoint Inhibitor	NCT05396833	I	Recruiting	Metastatic or locally advanced unresectable solid tumors
M4344 (VX803)	M4344; M4344 + Carboplatin	NCT02278250	I	Completed	Advanced solid tumors
	M4344 monotherapy; M4344 + Niraparib	NCT04655183	I/II	Withdrawn	Advanced solid tumors, breast cancer
	M4344 + Niraparib	NCT04149145	I	Not recruiting	PARPi-resistant recurrent OC
RP-3500	RP-3500; RP-3500 + Talazoparib + Gemcitabine	NCT04497116	I/II	Recruiting	Advanced solid tumors
	RP-3500; RP-3500 + RP-6306	NCT04855656	I	Recruiting	Advanced solid tumors
	RP-3500 + Niraparib and/or Olaparib	NCT04972110	I/II	Recruiting	Advanced solid tumors, adult

Clinical trial data are from: ICTRP Search Portal (who.int); clinicaltrials.gov; chictr.org.cn. Annotation: Avelumab: a human PD-L1 antibody; Copanlisib: a PI3K kinase inhibitor; Durvalumab: a human PD-L1 antibody; Etoposide: a podophyllotoxin derivative; Gemcitabine: a cytidine analogue; Irinotecan: a topoisomerase I inhibitor; Lurbinectedin: a Pol-II inhibitor; Niraparib: a PARPi; NSCLC: non-small cell lung cancer; Olaparib: a PARPi; Paclitaxel: the first microtubule-stabilizing agent; Pembrolizumab: a humanized monoclonal anti-PD1 antibody; RP-6306: an oral PKMYT1 inhibitor; SCC: squamous cell carcinoma; SCLC: small cell lung cancer; Talazoparib: a PARPi; Topotecan: a topoisomerase I inhibitor; Veliparib: a PARPi.

**Table 2 cancers-14-05902-t002:** Potential biomarkers of ATR inhibitors.

Regulatory Mechanism	Patient Selection
Blocked DNA synthesis and accumulation of damaged DNA	*ARID1A*-deficient [93,94]; *ATRX*-Deficient [95]; *BRG1* Loss (*SMARCA4*-deficient) [96,97,98]; LUC7L3-deficient [99]; *NEIL3*-deficient [100]; *POLE3/POLE4*-deficient [101]; *POLD1*-deficient [102,103]; PRIM1-deficient [104]; *REV3*-deficient [105]; *RNASEH2*-deficient [106]; *RAS*-transformed [107].
Impaired DNA damage repair (DDR)	HRD [108] (AXL [109]; BRCA [108,110], RAD51 [111], PARP14 [111]; *FANCM* [112,113]; *NEIL3*-deficient); *CCNE1* amplification [74]; DNA-PKcs-deficient [114]; ERCC1-deficient [115]; LIAS-deficient [99]; MED12 and PTEN-deficient [116]; MGMT-deficient [117,118]; XRCC1-deficient [119,120].
Stalling of replication fork progression	APOBEC3B reduced [121]; *BRG1* Loss (*SMARCA4*-deficient); *FANCM*-deficient; Loss of Cyclin C and CDK [122]; *MYCN* amplification [123]; *PBRM1*-defective [124]; PPP2R2A-deficient [125]; RAD51 reduced [111]; *REV3*-deficient; *SLFN11*-deficient [126,127]; TopBP1-deficient [128,129].
Regulation of the cell cycle	*ARID1A*-deficient; ATM-deficient; *CCNE1* amplification; DNA-PKcs-deficient; *FANCM*-deficient; KDM5D-defective [130]; NUSAP1-deficient [131]; Tim-Tipin-deficient [132]; *WWOX*-deficient [133].

Annotation: *ARID1A*: encodes a multifunctional BAF complex subunit that regulates transcription and recruits topoisomerase II to chromatin; AXL: a receptor tyrosine kinase involved in a cellular process; *BRG1*: *Brahma-related gene 1* (also known as *SMARCA4*) is a major factor in chromatin remodeling; *CCNE1*: encoded cyclin E1, which is an important contributor to the G1/S cell cycle transition; DNA-PKcs: DNA-dependent protein kinase catalytic subunit (DNA-PKcs) is a member of the PI3K family and plays an important role in multiple nodes of DDRs; ERCC1: a key regulator of the NER pathway; *FANCM*: encodes a multi-domain scaffolding and motor protein that interacts with several distinct repair protein complexes at stalled forks; KDM5D: a male-specific histone-modifying enzyme that represses certain genes at the level of transcription; MGMT: O^6^-methylguanine-DNA methyltransferase, is a DNA repair enzyme; *MYCN*: an oncogene; *NEIL3*: *Nei endonuclease VIII-like 3*; *ATRX*: encodes a SWI/SNF-like chromatin remodeling protein; *PBRM1*: an anti-oncogene encodes BAF180 protein; *POLD1*: encodes the catalytic subunit of polymerase (Pol) δ; *POLE3/POLE4*: encodes DNA polymerase ε accessory subunit; PPP2R2A: subunits of Protein phosphatase 2 which is a heterotrimeric serine/threonine (Ser/Thr); PRIM1: one of the DNA primase complexes; PTEN: negative regulator of PI3K; Ras: an oncogene; *REV3*: encodes catalytic subunit of transcription synthesizes polymerase ζ; *RNASEH2*: encodes the heterotrimeric RNaseH; *SLFN11*: one of *Schlafen* family genes; Tim-Tipin: associate with replisome components (MCM subunits, Pol δ/ϵ, and Claspin) and perform important functions in both DNA replication and genome maintenance; *WWOX*: encodes WW domain-containing oxidoreductase (a transcription regulator); XRCC1: X-ray cross-complementing group 1 protein that participates in a variety of DNA damage repair pathways.

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
