# Peer review of "ATR Inhibitors in Platinum-Resistant Ovarian Cancer"

_cancers, 2022, doi:10.3390/cancers14235902_

Round 1

Reviewer 1 Report

Li and colleagues presented a very comprehensive review on the therapeutic potential of ATRi in platinum-resistant ovarian cancer. Overall, the authors described all the relevant aspects related to the use of ATRi in PROC as well as the clinical trials active in this field. There are no major weaknesses, however, there are some minor issues that the authors have to address before publication:

1) Please revise the Title. Use only one of the two sentences;

2) Please check the error in the following paragraph: “It confirmed the radio sensitization of VE- 822. The study also showed that VE-822 did not show high toxicity to normal cells and tissues. It means that ATRi can promote the tolerated dose of normal tissues in the target area while reducing the lethal dose of tumor tissues, i.e., improving therapeutic ratio37. The study by Pang- Ning Teng et al. confirmed that VE-821 can enhance the reactivity to platinum drugs and the response of IR in several gynecologic cell lines, including OC cell lines.”. What is the correct name (VE-822 or VE-821)?

3) The authors described mainly the adoption of ATRi in PROC. What about the use of ATRi in other OCs? The authors have to better describe this aspect as, for example, the use of PARPi was first approved as a second-line treatment for BRCA-positive breast cancer while at present PARPi can be used also as first-line treatment or for the treatment of both BRCA-positive and negative OC. Please add also some information on the use of ATR in general;

4) I suggest to add a brief and separate chapter describing the current pharmacological treatments adopted for ovarian cancer (thus describing chemotherapy, targeted therapy, radiotherapy and immunotherapy);

5) Throughout the manuscript the authors described multidisciplinary findings which improved the therapeutic outcomes obtained in ovarian cancer patients. They also described potential biomarkers for the evaluation of patients’ responses. In the final part of the manuscript the authors have to add a brief description of the utility of a multidisciplinary approach for the treatment of ovarian cancer highlighting how specialized cancer centers where operate multidisciplinary teams obtain better outcomes in terms of patients’ survival and quality of life. For this purpose, please see:

- PMID: 34132354

- PMID: 35267603

- PMID: 33832445

- PMID: 34787913

Reviewer 2 Report

1. There are a lot of abbreviations in the articles, which makes them really hard to read. Please just use common abbreviations. It is unnecessary to have abbreviations like ATM inhibitor (ATMi), which only appeared four times, and Ovarian Cancer Consensus Conference (OCCC), which only appeared once.

2. Page 2: there is no definition of DSB, which has been mentioned six times throughout the manuscript. Please add it. 

3. Page 3: "RPA recruits ATR and ATR-interacting protein (ATRIP) to nuclear damage-induced nuclear foci, with DNA damage and stimulated ATRIP and ssDNA". Recruits to repair? I did not get the meaning of this sentence. 

4. The words in Figure 1 are hard to read. Please make them readable. 

5. Page 5: What is CTLA? Is Anti-CTLA-4 an antibody? Please use more explanation.

6. Similar to Figure 1, the content of table 1 is blurry and hard to read. Please revise it. 

7. Please keep the format matching the requirement of the publication. 

8. Page 9: It can inhibit the phosphorylating of CHK1 and increase the phosphorylation of γH2AX73. It should be "phosphorylation". Please also check your grammar. Please make sure there is no obvious grammar mistake.

9. Page 9: CAPRI is a Phase II trial in which 14 patients with platinum-resistant HGSOC were enrolled in cohort B to receive ceralasertib in combination with olaparib. Citation? Or what is CAPRI?

10. Page 11: I do not get the meaning of the paragraph. "About 50% of HGSOC patients have mutations in HRR-related genes, including BRCA1/2, RAD51, and ATM 102, 103. Nevertheless, the absence of these checkpoints opens up the possibility of using ATRi monotherapy. On the one hand, these markers provide a basis for the efficacy of ATRi in ovarian cancer. On the other hand, they also provide a basis for screening patients in clinical trials."

Reviewer 3 Report

In this manuscript, the Authors focused on a new class of anticancer agents called ATR inhibitors (ATRi) in the field of platinum-resistant ovarian cancer (PROC). They aimed to review the anticancer mechanism of ATRi and the progress of research on ATRi in PROC. 

The review provided a clear overall evaluation of ATRi. The Authors described in detail their mechanism of action, the preclinical data on ATRi-based combinations, and available data for these drugs in the clinical setting.

Here, I report my suggestions for the Authors: 

Major Revisions:

-Extensive editing of English language and style is required.

-I suggest adding a brief “Discussion” paragraph before the “Conclusion” paragraph, where the Authors summarize the data reported in the review, providing their perspectives and future directions for these drugs in the setting of platinum-resistant ovarian cancer. In addition, I suggest highlighting the emerging unmet need to overcome PARP inhibitors (PARPis) resistance, for whose condition ATRi could represent a promising strategy (Giudice E. et al, Cancers, 2022; Gupta et al, Pharmacol Res.  2022)

-ATRi and PARPis: the Authors should mention the involvement of the ATM/ATR pathway as one of the PARPis mechanisms of resistance; please broaden the discussion explaining the link between ATR and PARPis resistance. I suggest adding these two references (Giudice E. et al, Cancers, 2022; Gupta et al, Pharmacol Res.  2022) for this purpose.

-Table 1: Is this Table copied from another publication/book chapter? If not, I suggest reporting it according to the style required by the journal (the same style as Table 2). If yes, please follow the copyright instructions of the journal.

-Figure 1 is not mentioned in the text, and thus I suggest the Authors to mention it in the text. Please cite the data from which this figure is created. Finally, I suggest editing the Figure (e.g. texts are not centered in the circles)

-In the references list, it is written "!!! INVALID CITATION !!! 115." after reference No. 113. What does it mean?

Minor revisions:

-Introduction section: The following sentence should be revised: Nonetheless, a lot of patients either relapse within 6 months or don’t respond to the first-line chemotherapy and are called “platinum-re- sistant/refractory". This definition refers to PRIMARY platinum-resistant patients. As the authors correctly stated in the paragraph "What is PROC?", the correct definition of generic “platinum-resistant” is “recurrence or disease progression shorter than 6 months to LAST platinum-based chemotherapy”. Besides, patients are considered “platinum refractory” if the interval between the last dose of a platinum agent and the date of relapse is less than 1 month. Please clarify.

-Paragraph "What is PROC?" (Line 1): "patients" and not "Patients”;

-Paragraph “What is PROC?” (Line 8): “Platinum-free” and not “platinum-free”;

-Paragraph “What is PROC?” (Line 19): “progression-free” and not “Progression-free”;

-Paragraph “What is PROC?” (Line 20): “Response Evaluation Criteria in Solid Tumor (RECIST)” and not “RECIST (Response Evaluation Criteria in Solid Tumor)”;

-Paragraph “ATR/CHK1 pathway” (Line 22): The explicit meaning of PIKK should be added before the abbreviation;

-Figure 1: I suggest adding the explicit forms of all the abbreviations included in the figure;

-Paragraph “Combination with radiotherapy” (Line 20): “Teng et al.” and not “Pang- Ning Teng et al.” (I suggest removing the first name of the first author, leaving only the surname);

-Paragraph “Combination with chemotherapy” (Line 10): “Hall et al.” and not “Amy B Hall et al.”;

-Paragraph “Combination with immunotherapy” (Line 18): “Sun et al.” and not “Lin-Lin Sun et al.”;

-Paragraph “ATRi and Poly (ADP-Ribose) polymerase inhibitor (PARPi)” (Line 12): “Huntoon et al.” and not ”Catherine J. Huntoon et al.”;

-Paragraph “ATRi and Poly (ADP-Ribose) polymerase inhibitor (PARPi)” (Line 14): “Kim et al.” and not ” Hyoung Kim et al.”;

-Paragraph “ATRi and Topoisomerase inhibitors” (Line 11): “Josse et al.” and not “Rozenn Jossé et al.”;

-Paragraph “ATRi and Topoisomerase inhibitors” (Line 16): “Hur et al.” and not “Jin Hur et al.”;

-Paragraph “Molecular marker and determinants of Sensitivity of ATRi”: “Middleton et al.” and not “Fiona K. Middleton et al.”.

Round 2

Reviewer 3 Report

Authors addressed new angles to the manuscript, thus becoming very appealing and suitable for publication. Overall, quality has been improved due to authors’ editing.

However, just some few clarifications would be preferrable before manuscript publication.

Surgery in PROC does not represent a standard of care as well as immunotherapy. Secondary cytoreduction demonstrated an improvement in terms of PFS in the platinum-sensitive setting (DEKSTOP III trial), but not in the platinum-resistant setting. Concerning immunotherapy, several trials are ongoing combining immunotherapy to other agents such as PARPis, but thus far, this strategy represents only a promising valid alternative to standard chemotherapy.

Therefore, I would recommend editing the following sentences before the manuscript publication.

 “The treatment strategy for PROC is comprehensive treatment dominated by chemotherapy and targeted therapy supplemented by immunotherapy, surgery, and symptomatic therapy

 “Surgery is also an option for PROC and the surgical methods for PROC mainly include secondary cytoreductive surgery, intraperitoneal hyperthermic chemotherapy, and palliative surgery. A retrospective analysis showed that patients receiving secondary cytoreductive surgery plus chemotherapy had significantly longer postrelapse survival than patients treated with chemotherapy alone. The role of intraperitoneal hyperthermic chemotherapy is still controversial”.
